# What Are the Relationships between Psychosocial Community Characteristics and Dietary Behaviors in a Racially/Ethnically Diverse Urban Population in Los Angeles County?

**DOI:** 10.3390/ijerph18189868

**Published:** 2021-09-19

**Authors:** Brenda Robles, Tony Kuo, Courtney S. Thomas Tobin

**Affiliations:** 1Department of Community Health Sciences, UCLA Fielding School of Public Health, P.O. Box 951722, Los Angeles, CA 90095, USA; courtneysthomas@ucla.edu; 2Department of Epidemiology, UCLA Fielding School of Public Health, P.O. Box 951722, Los Angeles, CA 90095, USA; tkuo@mednet.ucla.edu; 3Department of Family Medicine, David Geffen School of Medicine at UCLA, 10880 Wilshire Blvd, Suite 1800, Los Angeles, CA 90024, USA; 4Population Health Program, UCLA Clinical and Translational Science Institute, 10833 Le Conte Ave, BE-144 CHS, Los Angeles, CA 90095, USA

**Keywords:** public health interventions, policy, systems and environmental changes, psychosocial community characteristics, chronic disease prevention, fruit and vegetable consumption, soda consumption

## Abstract

To address existing gaps in public health practice, we used data from a 2014 internet panel survey of 954 Los Angeles County adults to investigate the relationships between psychosocial community characteristics (PCCs) and two key chronic disease-related dietary behaviors: fruit and vegetable (F+V) and soda consumption. Negative binomial regression models estimated the associations between ‘neighborhood risks and resources’ and ‘sense of community’ factors for each dietary outcome of interest. While high perceived neighborhood violence (*p* < 0.001) and perceived community-level collective efficacy (*p* < 0.001) were associated with higher F+V consumption, no PCCs were directly associated with soda consumption overall. However, moderation analyses by race/ethnicity showed a more varied pattern. High perceived violence was associated with lower F+V consumption among White and Asian/Native Hawaiian/Other Pacific Islander (ANHOPI) groups (*p* < 0.01). Inadequate park access and walking as the primary mode of transportation to the grocery store were associated with higher soda consumption among the ANHOPI group only (*p* < 0.05). Study findings suggest that current and future chronic disease prevention efforts should consider how social and psychological dynamics of communities influence dietary behaviors, especially among racially/ethnically diverse groups in urban settings. Intervention design and implementation planning could benefit from and be optimized based on these considerations.

## 1. Introduction

Over the past decade, there has been growing recognition that the built environment and the context of food at the community/neighborhood level can influence how and what people eat [1,2]. The dietary decision-making process is known by many as a modifiable determinant of chronic disease risk [3]. As such, in the United States (U.S.), policy, systems, and environmental change interventions (PSEs) since 2010 have promoted healthy eating, largely by intervening within built and food environments [4,5]. While the built environment is defined as “the human-made space in which people live, work, and recreate on a day-to-day basis” [6] (p. 25), food environments are known as “the collective physical, economic, policy and socio-cultural surroundings, opportunities, and conditions that influence people’s food and beverage choices and nutritional status” [7] (p. 25).

Although PSEs are often guided by the Social Ecological Model (i.e., a framework that emphasizes the interactive effects of multiple determinants of health across several ecological levels) [8], they have principally focused on making structural and institutional changes to promote healthy dietary decisions among individuals [4,5,9]. Consequently, such PSEs frequently overlook other ecological factors that may have important implications for designing or tailoring evidence-based, nutrition-focused public health interventions for diverse populations [9,10]. Factors that may be particularly influential include community contexts, such as level of neighborhood social cohesion or social capital [11], and psychosocial factors, such as depression, psychological distress, and/or other mental health conditions [12,13].

Emerging evidence suggests that these contexts and characteristics can also determine how and what people eat. For example, prior research has demonstrated the significant impact of neighborhood determinants (e.g., neighborhood violence, etc.) [14] and neighborhood resources (e.g., park access, store distance, mode of transportation, etc.) on diet [15,16,17,18]. Additional factors, such as a community’s collective efficacy and degree of economic hardship, have also been found to attenuate or accentuate health outcomes, including cardiovascular disease risk [15,19,20,21,22] and life expectancy/mortality [23,24,25]. There is also emerging evidence that neighborhood satisfaction matters for health and well-being [26].

These and other factors, hereon collectively referred to as *psychosocial community characteristics (PCCs)*, remain less understood and understudied in the field of chronic disease prevention and control. The construct of PCCs is, in part, derived from a growing body of research that implicates race and racism as fundamental determinants of health [27,28], including their impact on diet-related outcomes [29,30]. This scholarship also suggests that access to community-level resources and PCCs may vary significantly across various racial/ethnic groups due to historical and contemporary racism (e.g., redlining and its aftereffects, etc.) [31,32]. Since previous studies also indicate that racial differences exist, in community/neighborhood environments [32,33,34], as well as health status [29,30], it is possible that the impact of PCCs on diet may also vary by race/ethnicity. Therefore, if the question of whether race/ethnicity moderates the relationships between PCCs and dietary decisions can be answered, a more granular approach to designing or tailoring public health interventions to fit the needs of target populations could become a more common practice and cost-effective strategy for addressing health equity.

The present study sought to close some of these knowledge gaps in public health practice by answering the following research question: What are the relationships between PCCs and dietary behaviors in a racially/ethnically diverse urban population in Los Angeles County (LAC)? The study’s specific aims were: (1) to describe the relationships between PCCs and the two dietary behavioral outcomes of interest: fruit and vegetable [F+V] and soda consumption; and (2) to examine whether race/ethnicity moderates these relationships. We focused on these two dietary outcomes because enhancing the consumption of F+Vs and limiting sugar-sweetened beverages such as soda are often the focus of diet-related chronic disease prevention efforts. Although such standards are often delineated in national dietary recommendations [35], few studies have explored group differences in the ways that community contexts influence individuals’ ability to adhere to these standards.

## 2. Materials and Methods

### 2.1. Conceptual Framework

We used the Social Ecological Model as a framework because it delineates how various factors at different ecological levels (i.e., public policies, communities, institutional, and inter- and intra-personal) interact with one another to shape health behaviors [8]. Yet, while practical, this model does not always clearly differentiate or show how sociological, psychological, and biological factors behave individually or in combination across different ecological levels to shape health outcomes or racial/ethnic health disparities at the community-level. Thus, to complement the strengths of the Social Ecological Model, other perspectives were integrated to develop the present study’s conceptual framework (see Figure 1). Namely, we considered the Biopsychosocial Model and Environmental Affordances Model, as these additional models enhance many of the constructs and pathways described by the Social Ecological Model. In general, the Biopsychosocial Model better acknowledges how sociological, psychological, and biological factors interact with each other and in combination to influence health [36]. Similarly, the Environmental Affordances Model offers more nuanced explanations of the potential intersections between stress, health behaviors, and physical and mental health, including the dynamic pathways through which stress and social contexts shape health behaviors by population diversity [37].

Combining these various models into the study’s conceptual framework helped to shed light on the potential racial/ethnic differences that may exist for the relationships between PCCs and chronic disease-related dietary behaviors. This framework also allowed us to propose and chart two domains of PCCs that are believed to affect dietary behaviors: (a) exposure to neighborhood risks and resources; and (b) factors that contribute to a ‘sense of community’. We define *neighborhood risks and resources* as individuals’ perceived access to physical environments or resources that may help or hinder efforts towards healthy eating, and *sense of community* factors as how socially connected and happy individuals feel about and within their community.

### 2.2. Internet Panel Survey

The study utilized data from a cross-sectional internet panel survey conducted by a California-based survey firm contracted by the Los Angeles County Department of Public Health (DPH). This firm invited survey participants from their proprietary panel of nearly 14 million global subscribers to take a ~20 min, web-based survey. Due to the limited capacity to translate the questionnaire, the survey was only made available in English. The firm collected data at a single time point in 2014 (10 October to 15 November). To be eligible, subscribers from the firm’s global panel had to be a resident of LAC and ≥18 years of age. Prospective survey participants were also prompted to answer sociodemographic questions that asked them to indicate their sex, age, race/ethnicity, income, and education level. This sociodemographic information was used to determine and apply quota targets to the sample so that they aligned with the 2010 U.S. Census estimates for adults in the region (i.e., the firm sampled from eligible survey panel subscribers until saturation was reached for the sex, age, race/ethnicity, income, and education level stratum). This application of U.S. Census-based quota criteria has been previously employed in similar surveys as a way to improve a sample’s representativeness of the LAC population [38,39].

To recruit survey participants, eligible panel subscribers (i.e., LAC resident, ≥18 years of age, and who matched sociodemographic quota targets) were sent an initial email inviting them to be part of the survey. Those who did not respond to this initial email were later sent two follow-up reminders. Subscribers who clicked on the survey link in the invitation email(s) were logged into their unique “dashboard” where the survey administration then took place. Only subscribers who completed 70–100% of the questions were considered enrolled in the study. They received dashboard points equivalent to a monetary incentive of $2.25 for finishing the online questionnaire. With a total of 1000 subscribers (participants) who completed the survey, the participation rate was estimated to be about 33%. Survey methods and protocols, including the application of target quotas, have been described in more detail elsewhere [38,39].

In the present study, 13 participants who self-identified as “American Indian/Alaskan Native” or “Other” were dropped from the analysis due to small cell sizes. In addition, only those who provided complete information for the variables of interest were included in the study analyses. The final analytic sample was 954.

### 2.3. Dependent Variables

#### 2.3.1. F+V Consumption

The following questions, adapted from the validated Diet History Questionnaire from the Eating at America’s Table Study [40], were used to assess survey participants’ average daily F+V consumption: (1) “In an average day, about how many servings of fruit do you eat, counting fresh, canned, dried, or frozen fruits? A serving is defined as the following: (a) 1 medium fruit (such as apples, oranges, bananas, pears); (b) ½ cup chopped, cooked, or chopped, fruit; or (c) ¾ cup fruit juice”; and (2) “In an average day, about how many servings of vegetables do you eat, counting fresh, canned, dried, and frozen vegetables. A serving is defined as the following: (a) 1 cup of raw leafy vegetables (such as lettuce); (b) ½ cup of other vegetables (either cooked, raw, or chopped); or (c) ¾ cup of vegetable juice.” For both questions, participants reported their answers as whole-number values (i.e., counts). These values were then summed, but with implausible counts (i.e., >16 for fruits and >23 for vegetables) excluded from the analyses based on national data for adult F+V consumption [41]. In the descriptive analyses, F+V consumption was measured categorically: (0) optimal consumption (≥5 servings of F+Vs per day); (1) intermediate consumption (3–4 servings of F+Vs per day); or (2) worse consumption (0–2 servings of F+Vs per day). These cut-points were based on prior national recommendations and campaigns encouraging individuals to consume five or more F+Vs daily for optimal health [42,43,44]. In the multivariable regression analyses, daily F+V consumption was analyzed as a count.

#### 2.3.2. Soda Consumption

Weekly soda consumption was assessed by asking participants the following question used in prior DPH studies [38,45]: “In an average week, about how many regular sodas such as Coke or Mountain Dew, do you drink? Do not include diet sodas or sugar-free drinks. Please count a 12-ounce can, bottle or glass as one drink”. Study participants were asked to report their responses as whole-number values. Implausible values (i.e., >22 sodas or more per day) were excluded from the analyses, which aligns with cut-points and the approach employed in a previous study [46]. In the descriptive analyses, soda consumption scores were categorically measured: (0) optimal consumption (0 sodas per week); (1) intermediate consumption (1–6 sodas per week); or (2) worse consumption (7 or more sodas per week). These categories are consistent with those employed by other studies in LAC [38,47]. Like daily F+V consumption, soda consumption was analyzed as a count in the regression analyses.

### 2.4. Independent Variables

#### 2.4.1. Neighborhood Risks and Resources

Perceived neighborhood violence: Perceived neighborhood violence was measured with a 5-item scale (α = 0.92) adapted from the Project on Human Development in Chicago Neighborhoods Community Survey [48]. Survey participants reported the level to which the following events took place in their neighborhood during the past six months: (a) fight involving a weapon; (b) violent argument between neighbors; (c) gang fight; (d) sexual assault; (e) robbery/mugging; and (f) police officer harassment/abuse/unjustified use of force. Response options ranged from “often” (coded as 5) to “never” (coded as 1). Items were summed to create a continuous score where higher values indicate a higher level of perceived neighborhood violence. To compare risk groups more easily in study analyses, we categorized perceived neighborhood violence into terciles: (0) low violence; (1) intermediate violence; and (2) high violence.

Park access: Participants were asked to indicate “yes” or “no” to the following question: “Is there a park, playground, or open space within walking distance of your home?” This question was adapted from the California Health Interview Survey [49]. In study analyses, responses were dichotomized as: (0) yes, has park access; and (1) no, does not have park access.

Grocery store distance: Participants were asked, “Approximately how far is the place you generally get most of your groceries (in miles).” Response options were reported as whole numbers and were analyzed as a continuous variable in the study analyses. This question was adapted from prior DPH studies [50,51].

Mode of transportation: Participants were asked, “What mode of transportation do you usually take [to get to the place where you usually get most of your groceries]?” Responses were categorized as follows in the study analyses: (0) car [car/rideshare]; (1) bus; (2) walking; or (3) other [biking, other]. This measure was similarly operationalized in previous studies of the LAC population [50,51].

Community-level economic hardship: Community-level economic hardship was assessed using the LAC 2008–2012 Economic Hardship Index, which represents a composite score of crowded housing, unemployment, education, dependency, and per capita income. Scores ranged from 13.2 to 82.5, with higher values indicating greater community-level economic hardship. Prior studies using the same dataset as in the present study described how the community-level economic hardship variable was constructed [38,39].

#### 2.4.2. Sense of Community

Perceived community-level collective efficacy: This captures the extent to which individuals feel connected to their neighbors and perceive their neighbors as willing to intervene on the behalf of the common good [52]. This variable utilizes measures of social cohesion and informal social control adapted from the Project on Human Development in Chicago Neighborhoods Community Survey [48], which is based on prior research [52,53].

Perceived social cohesion was measured with a 5-item scale (α = 0.56), based on a series of questions asking participants to indicate the extent to which they consider: (a) their neighborhood as close-knit or unified; (b) people in their neighborhood as willing to help other neighbors; (c) people in their neighborhood as not getting along with other neighbors; (d) people in their neighborhood as not sharing the same values; and (e) people in their neighborhood as trustworthy. Response options ranged from (1) “strongly agree” to (5) “strongly disagree” for “negative” items (i.e., indicative of adverse community interactions); “positive” items (i.e., indicative of positive community interactions) were reverse coded, such that when summed, higher scores corresponded with higher levels of perceived social cohesion.

Perceived informal social control was also measured with a 5-item scale (α = 0.82). Participants were asked a series of questions about the level of disorder in their neighborhood: (a) “If a group of neighborhood children were skipping school and hanging out on a street corner, how likely is it that your neighbors would do something about it?”; (b) “If some children were spray-painting graffiti on a local building, how likely is it that your neighbors would do something about it?”; (c) “If a child was showing disrespect to an adult, how likely is it that people in your neighborhood would scold that child?”; (d) “If there was a fight in front of your house or building and someone was being beaten or threatened, how likely is it that your neighbors would break it up?”; and (e) “Suppose that because of budget cuts the fire station closest to your home was going to be closed down by the city. How likely is it that neighborhood residents would organize to try to do something to keep the fire station open?” Response options ranged from “very likely” (coded as 5) to “very unlikely” (coded as 1).

For each of the perceived social cohesion and perceived informal social control items, responses were summed to create a continuous score where higher values indicated a higher level. Consistent with the approach employed by Sampson and colleagues (1997), a factor analysis confirmed that the two scales could be combined into a single measure to represent a single latent construct, known as “collective efficacy” [52]. Thus, responses from both scales were summed to create a composite score. These scores were assessed as a continuous variable in descriptive and regression analyses, with higher scores indicating higher levels of perceived community-level collective efficacy.

Neighborhood satisfaction: Participants were asked to indicate their level of satisfaction with their neighborhood: “All things considered, would you say you are very satisfied, satisfied, dissatisfied, or very dissatisfied with your neighborhood as a place to live?” Responses were categorized as follows in the study analyses: (0) satisfied [very satisfied/satisfied]; or (1) unsatisfied [dissatisfied, very dissatisfied]. This measure was adapted from a question previously used in the Los Angeles Family and Neighborhood Survey (L.A. FANS) [54].

### 2.5. Covariates

For the study analyses, several self-reported sociodemographic characteristics were included as covariates, with the largest category selected as the reference group for each variable examined: sex (categorized as 0 = male; 1 = female); age (categorized as 0 = 18–30 years; 1 = 31–40 years; 2 = 41–50 years; and 3 = greater than 50 years in descriptive analyses; analyzed as a continuous variable in regression analyses); race/ethnicity (categorized as 0 = Hispanic; 1 = Black; 2 = White; and 3 = Asian/Native Hawaiian/Other Pacific Islander [ANHOPI)]); nativity status (categorized as 0 = born in LAC; 1 = native born but outside of LAC; 2 = foreign born); language spoken at home (categorized as 0 = English; 1 = not English); education (categorized as 0 = college graduate/postgraduate; 1 = high school education or less; and 2 = technical/vocational school or some college); employment status (categorized as 0 = employed full-time; 1 = employed part-time; 2 = unemployed; and 3 = other employment status); income (categorized as 0 = less than $50,000; 1 = $50,000–$99,000; and 2 = over $100,000); marital status (categorized as 0 = single/never married; 1 = married; and 2 = divorced/separated/widowed); and children in the household (reported as whole-number values and analyzed as a continuous variable).

### 2.6. Statistical Analyses

Univariate distributions of all study outcomes, predictors, and covariates were first examined using histograms, frequency/percentage measures, central tendency measures (e.g., means, median), and dispersion measures (e.g., range, standard deviation). These analyses informed variable selection and appropriateness of statistical procedures in the subsequent multivariable regression analyses. Guided by the study’s conceptual framework (Figure 1), Study Aim 1 investigated the extent to which PCCs were associated with each of the dietary outcomes of interest: F+V and soda consumption. Since both dietary outcomes were counts and over-dispersed, we used negative binomial regression models with robust standard errors to generate incidence rate ratios (IRRs) and 95% confidence intervals (CIs). Model 1 examined the relationships between PCCs and each outcome. Sociodemographic characteristics were added in Model 2. For Study Aim 2, we examined whether race/ethnicity moderated the relationships between PCCs and dietary behaviors by testing interactions between each PCC and race/ethnicity category for each of the dietary outcomes. All data were analyzed using the statistical software, STATA version 14.1 (StataCorp LP, College Station, TX, USA).

## 3. Results

### 3.1. Descriptive Analyses

Participant characteristics are presented in Table 1. Mean F+V and soda consumption was 5.6 servings per day and 4.4 sodas per week, respectively. Over one-third of the survey participants perceived low levels of neighborhood violence (38.2%), while a majority reported having access to a park (81.1%) and using a car as their primary mode of transportation to get to the nearest grocery store (81.2%). The mean grocery store distance was 4.1 miles and the mean community economic hardship score was 50.4. For the *sense of community* factors, the mean perceived community-level collective efficacy was 33.8. About 89.3% of the participants reported being very satisfied/satisfied with their neighborhood. Only slightly more than half of the participants were male (50.5%). The majority reported that they were between age 18–30 years (42.0%), Hispanic (49.1%), and born in LAC (66.8%). The majority also reported speaking English as the primary language at home (76.3%), having attained a college/postgraduate degree (44.8%), being employed full-time (55.7%), having an income under $50,000 (45.7%), and having a ‘single’ marital status (53.6%). The mean number of children per household was 0.8.

### 3.2. Negative Binomial Regression Analyses

#### 3.2.1. F+V Consumption

Results from the negative binomial regression analyses are presented in Table 2. Among survey participants, those who reported high perceived neighborhood violence consumed more F+Vs per day than those who reported low perceived neighborhood violence in the PCCs only and full model (IRR = 1.27, 95% CI = 1.13–1.43 in the PCCs only model; IRR = 1.24, 95% CI = 1.11–1.40 in the full model). Perceived community-level collective efficacy was also associated with higher F+V consumption in the PCCs only model (IRR = 1.02, 95% CI = 1.01–1.02) and the full model (IRR = 1.02, 95% CI = 1.01–1.02). Grocery store distance was marginally associated with F+V consumption in both the PCCs only model (IRR = 1.01, 95% CI = 1.00–1.02) and full model (IRR = 1.01, 95% CI = 1.00–1.02). ‘Other’ mode of transportation to the nearest grocery store was associated with F+V consumption in the PCCs only model (IRR = 1.45, 95% CI = 1.02–2.06), but this result was not statistically significant in the full model.

Several sociodemographic characteristics were found to be associated with daily F+V consumption (data not shown in Table 2). Compared to participants between the ages of 18–30, those who were 31–40 years old, 41–50 years old, and over the age of 50 consumed fewer F+Vs per day (31–40 years old: IRR = 0.85, 95% CI = 0.76–0.96; 41–50 years old: IRR = 0.78, 95% CI = 0.68–0.91; over 50 years of age: IRR = 0.80, 95% CI = 0.67–0.96). Participants born in the U.S. (but born outside of LAC) also consumed fewer F+Vs per day than their counterparts who were born in LAC (IRR = 0.86, 95% CI = 0.76–0.97). In contrast, those who were born outside of the U.S. consumed more F+Vs per day than those born in LAC (IRR = 1.19, 95% CI = 1.02–1.39).

#### 3.2.2. Soda Consumption

Survey participants who reported high perceived neighborhood violence drank more sodas per week than those who reported low perceived neighborhood violence in the PCCs only model (IRR = 1.42, 95% CI = 1.14–1.77). Grocery store distance was also found to be associated with soda consumption in the PCCs only model (IRR = 1.02, 95% CI = 1.01–1.03). However, after adjusting for sociodemographic characteristics, these associations were no longer statistically significant.

Several sociodemographic characteristics were found to be associated with soda consumption (data not shown in Table 2). Participants over the age of 50 drank fewer sodas per week than those between the ages of 18–30 (IRR = 0.56, 95% CI = 0.41–0.78). Compared to participants who self-identified as Hispanic, those who self-identified as ANHOPI drank fewer sodas per week (IRR = 0.62, 95% CI = 0.47–0.82). There was also lower soda consumption per week among participants who did not speak English versus those who spoke English as the primary language at home (IRR = 0.78, 95% CI = 0.64–0.96). Participants who had a high school education or less drank more sodas per week than those with a college/postgraduate education (IRR = 1.37, 95% CI = 1.04–1.79). Additionally, participants who were employed part-time drank fewer sodas per week than those who were employed full-time (IRR = 0.74, 95% CI = 0.57–0.96).

### 3.3. Moderation Analysis

There were several significant results from the moderation analysis, which explored the moderating effect(s) of race/ethnicity on the associations between PCCs and the two dietary behaviors of interest. These relationships are depicted in Figure 2, Figure 3 and Figure 4. When exposed to high levels of violence, White and ANHOPI participants consumed fewer servings of F+Vs per day than Hispanic participants (White: 0.71, 95% CI = 0.55–0.92; ANHOPI: 0.64, 95% CI = 0.47–0.87); there were no statistically significant differences in daily F+V consumption among Black and Hispanic groups.

Another significant relationship was the moderating effects of race/ethnicity on the associations between perceived park access and weekly soda consumption (Figure 3). Specifically, ANHOPI participants who reported inadequate park access consumed more sodas per week than Hispanic participants who reported inadequate park access (CI = 2.51, 95% CI = 1.20–5.28).

Race/ethnicity also significantly moderated the effects of mode of transportation to the nearest grocery store and weekly soda consumption (Figure 4). ANHOPI participants who reported walking as their primary mode of transportation to the nearest grocery store consumed more sodas per week than their Hispanic counterparts (IRR = 3.09, 95% CI = 1.06–9.00). And while White participants who used the bus as their primary mode of transportation to the nearest grocery store appeared to consume more sodas per week, this pattern was not statistically significant.

## 4. Discussion

The present study is among the first to conceptualize multiple PCCs as nonconventional but strong influencers of F+V and soda consumption, two dietary behaviors that are linked to chronic disease risk. The study is also among the first to describe these characteristics for a racially/ethnically diverse urban population in the U.S. Several notable findings can be derived from the study analyses, including that only certain PCCs are predictive of the two dietary behaviors of interest.

### 4.1. F+V Consumption

High perceived neighborhood violence, grocery store distance, and perceived community-level collective efficacy were associated or marginally associated with F+V consumption. These associations persisted even after adjusting for participant sociodemographic characteristics. While the positive linear relationship between perceived community-level collective efficacy and F+V consumption aligns with findings from a prior research study [55], the associations between high violence and greater distance to the nearest grocery store and higher F+V consumption are less intuitive. Prior studies suggest that neighborhood violence puts individuals at increased risk for obesity [56], a health outcome linked to F+V consumption [57]. Increased grocery store distance has also been implicated with lower F+V consumption [58]; and in LAC, null associations between grocery store distance and F+V consumption have been observed [50].

The potential moderating effect that race/ethnicity has on the associations between these two PCCs (i.e., perceived neighborhood violence and grocery store distance) and F+V consumption may explain some of the surprising results. There is some evidence that racial/ethnic differences exist in the relationship between grocery store distance and F+V consumption. For example, a prior study found an inverse relationship between miles to a grocery store/supermarket and daily F+V servings among Blacks, but not Whites [59]. And while few studies have explicitly investigated the linkages between perceived neighborhood violence and F+V consumption by race/ethnicity, moderation analysis results in the present study support PCCs, such as perceived neighborhood violence, as having a differential effect on dietary behaviors of distinct racial/ethnic groups. In particular, high perceived neighborhood violence appears to be more closely related to lower F+V consumption among White and ANHOPI versus Black and Hispanic groups. This finding somewhat aligns with a recent study that found food insecurity was associated with worse diet quality among non-Hispanic White, Asian, and “Other” racial/ethnic groups [60].

There are several plausible explanations for why perceived neighborhood violence may differentially impact the F+V consumption of distinct racial/ethnic groups. First, although Black and Hispanic residents tend to live in less affluent areas [61], the result that perceived neighborhood violence is not significantly associated with daily F+V consumption of Black and Hispanic groups could be because, regardless of socioeconomic status [62,63], F+Vs are likely still accessible in these low-income neighborhoods and their adjacent communities. For instance, some research has found no significant differences in the price of healthy foods sold in supermarkets by neighborhood socioeconomic status and Black/Hispanic neighborhood composition [64]. In addition, investments to combat obesity in LAC during the last decade likely contributed to increases in F+V consumption among Black and Hispanic populations, given these interventions primarily targeted low-income areas where many of these groups reside. Since 2010, these program efforts have included (but are not limited to) corner store conversions [65] and related supply chain strategies [66], increased acceptance of Electronic Benefit Transfer at farmers markets [51], multi-component faith-based interventions [67], and implementation of healthy food procurement policies in several organizational/institutional settings [68]. 

Second, the racial/ethnic differences observed in the impact of perceived neighborhood violence on F+V consumption (i.e., White and ANHOPI groups appear to consume fewer servings of F+Vs per day when they perceive high neighborhood violence, whereas high perceived neighborhood violence does not appear to significantly influence the F+V consumption behaviors of Black and Hispanic groups) could be explained, in part, by how these different groups react to stress. Prior research suggests that exposure to stress (both perceived and measured) is associated with a greater drive to eat among Black and Hispanic populations when compared with other racial/ethnic groups [69]. For example, trauma exposure and distress have been found to increase susceptibility to binge eating among Black trauma survivors [70] and binge eating has been noted as a problem among Black women [71]. Other studies suggest that Black men frequently respond to stress by engaging in an abundance of both healthy and unhealthy coping behaviors [72], including consuming larger portions of food. Similar findings have been described for Hispanic groups as well [73]; these findings could, in part, be explained by exposure to discrimination [74]. 

Third, cultural-specific dietary preferences of Black and Hispanic groups may play a role in explaining why high violence was not associated with F+V consumption. Traditional cuisines of both Black and Hispanic groups, for instance, include a variety of F+V options [75,76], suggesting that high F+V consumption should probably be expected in this research. Acculturation may, in part, also explain the relationship between violence and F+V consumption among Black and Hispanic groups. There is research evidence to support the observation that first-generation populations typically maintain healthier diets than second and third generations [77], and this may be the case for Hispanic and Black immigrants in LAC.

Finally, it should be noted that the mean number of F+Vs consumed per day was relatively high in this study sample. This was likely due to the fact that, at the time of data collection, several chronic disease-related PSEs (including those focused on promoting F+V consumption) had already been implemented in the LAC region. That the survey participants were predominately Hispanic and between the ages of 31–50 offer other potential explanations for the high F+V consumption per day, as these groups have historically been found to adhere to F+V dietary recommendations, more so than other groups [78]. Prior research also suggests that F+V consumption is generally higher in LAC as compared to other jurisdictions. A study comparing F+V intake among low-income priority populations in LAC and San Francisco revealed that LAC study participants consumed more F+Vs than their counterparts in San Francisco [79].

### 4.2. Soda Consumption

High perceived neighborhood violence and grocery store distance were associated with higher weekly soda consumption, but these relationships disappeared after adjusting for participant sociodemographic characteristics. A likely explanation for this null association could be the ubiquitous nature of soda consumption around the time that the present study was conducted—i.e., high levels of soda consumption already existed across all groups in LAC (likely because this sugar-sweetened beverage is highly affordable and accessible). Results from the Los Angeles County Health Survey, for example, showed that in 2015 more than two-thirds (75.5%) of Angelenos reported drinking one or more soda(s) per week; about one-third reported drinking one or more soda(s) per day [80]. Previously, to address this high prevalence of soda consumption, a health marketing communication campaign was launched and disseminated by DPH in 2011 to reduce this behavior among children and adults residing in LAC [81].

While no significant associations between PCCs and soda consumption were found after controlling for sociodemographic characteristics, there were some significant interactions by race/ethnicity. Specifically, significant racial/ethnic differences were found for weekly soda consumption and its associations with park access and mode of transportation to the nearest grocery store. ANHOPI participants with inadequate park access reported higher soda consumption as did ANHOPI participants who reported walking to the nearest grocery store as a primary mode of transportation. Several plausible explanations likely apply to these findings, including that unhealthy food, such as soda, typically have a longer shelf life than healthier food items, suggesting they are more available to Angelenos over a longer period. 

### 4.3. Study Limitations

The overall design and implementation of the present study are subject to some limitations. First, the data were self-reported, which could have introduced social desirability and recall bias. However, efforts were made throughout the study to mitigate these biases (i.e., examples of F+V serving sizes were provided as part of the survey administration process). Second, some survey measures were internally developed, raising questions about their reliability and validity. Nonetheless, these questions posed limited threats to the questionnaire design, as many of the items were adapted from previously validated survey questions or were pretested whenever possible. Third, the survey was only administered in English, which may have reduced the diversity of the study sample; albeit U.S. Census-based sociodemographic targets (quotas) were applied to improve the representativeness of the sample. Fourth, the study examined only a small subset of chronic disease and diet-related behaviors (i.e., F+V and soda consumption). Other behaviors such as those related to salt/sodium intake could have offered even more insights into dietary disparities and chronic disease risk in LAC [82]. Finally, calculating a true response rate was challenging given the general nature of internet panel survey data collection methods. In its place, a participation rate was estimated (i.e., ~33%), which, in comparison, was similar to rates from other cross-sectional surveys conducted previously in LAC [83].

Despite these limitations, the present study has strengths and is unique in several ways. For one, it is among the first to conceptualize and examine how social and psychological dynamics of communities may shape the dietary decisions of hard-to-reach, racially/ethnically diverse populations in LAC, with different data collected and analyzed within a single study. This research study also provides new insights into how PCCs may interact with sociodemographic characteristics to influence F+V and soda consumption, two key health behaviors that are known to reduce or increase chronic disease risk, especially for racially minoritized populations and/or those that live in under-resourced communities.

## 5. Conclusions

The present study bolsters the current knowledge base regarding factors that may influence healthy eating, as it describes the complex but important relationships between PCCs and two common measures of nutrition and chronic disease risk in the U.S. These results come at an opportune time, especially as the nation continues to grapple with how best to prevent and control chronic disease conditions such as obesity, diabetes, and hypertension – i.e., risk factors for heart disease and stroke. Recent studies have also demonstrated that these diet-related conditions may amplify severe infection and risk of death from the novel coronavirus disease 2019 [84,85]. Now, and more than ever, facilitating healthy eating has become a critical and strategic arsenal for a broader public health strategy to reduce chronic disease risk. From a policy and health equity standpoint, these results point to an urgent need to consider and incorporate PCCs into PSE designs. The approach will become ever more important to the future development and implementation of public health interventions, especially for those that are intended to inform policymaking at all levels (federal, state, local) and to improve nutrition programming for racially/ethnically diverse urban populations across the U.S.

## Figures and Tables

**Figure 1 ijerph-18-09868-f001:**
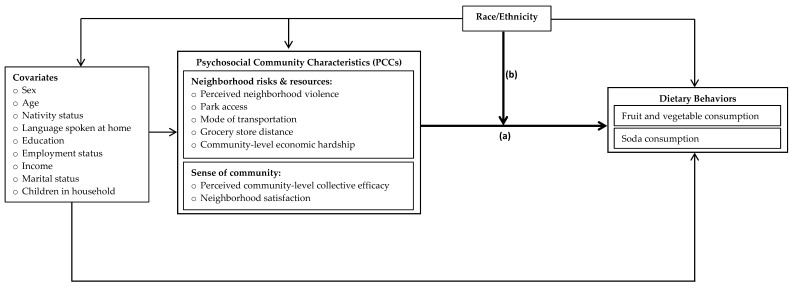
Conceptual framework of proposed relationships between psychosocial community characteristics (PCCs) and dietary behaviors and the potential moderating role of race/ethnicity. Note: the present study examines the bolded lines in the framework—(a) Study Aim 1: associations between PCCs and dietary behaviors; and (b) Study Aim 2: the moderating effects of race/ethnicity on the PCC-dietary behaviors relationships.

**Figure 2 ijerph-18-09868-f002:**
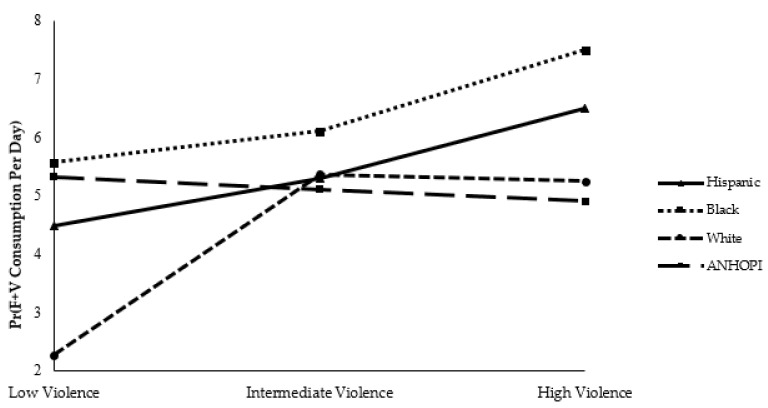
The association between perceived neighborhood violence and fruit and vegetable [F+V] consumption is moderated by race/ethnicity.

**Figure 3 ijerph-18-09868-f003:**
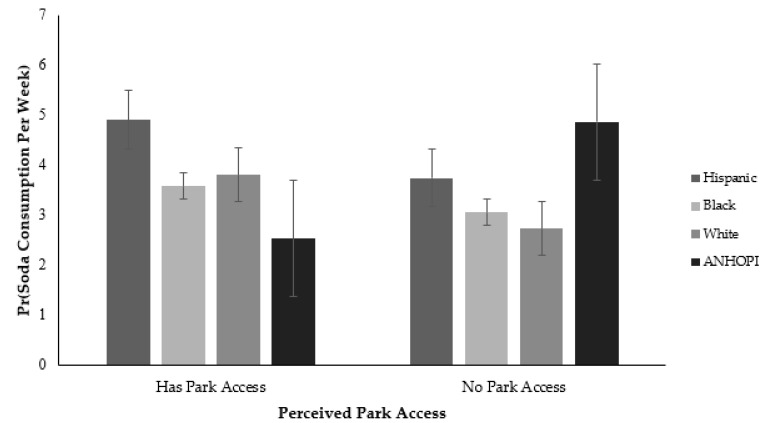
The association between perceived park access and soda consumption is moderated by race/ethnicity.

**Figure 4 ijerph-18-09868-f004:**
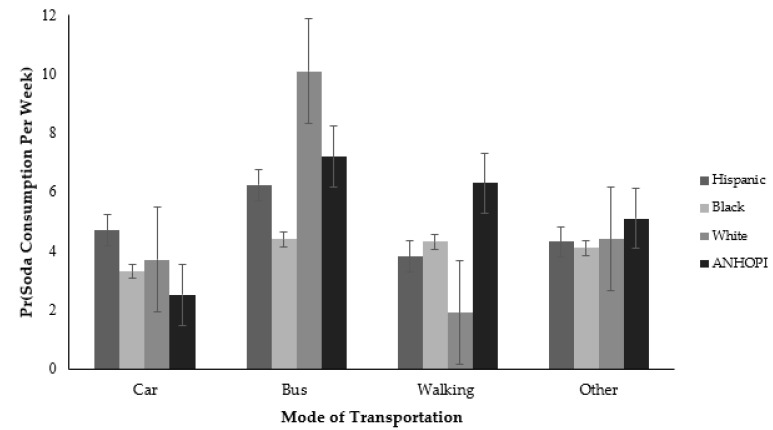
The association between mode of transportation to the nearest grocery store and soda consumption is moderated by race/ethnicity.

**Table 1 ijerph-18-09868-t001:** Survey participant dietary behaviors, psychosocial community characteristics, and sociodemographic characteristics (*n* = 954) ^a^.

Characteristics	Number (%) or *Mean [SD]*
**Dietary Behaviors**	
Fruit and vegetable consumption	
Optimal consumption (≥5 servings per day)	472 (49.5)
Intermediate consumption (3–4 servings per day)	278 (29.1)
Worse consumption (0–2 servings per day)	204 (21.4)
*Mean fruit and vegetable consumption*	*5.6 [4.3]*
Soda consumption	
Optimal consumption (0 sodas per week)	233 (24.4)
Intermediate consumption (1–6 sodas per week)	532 (55.8)
Worse consumption (≥7 sodas per week)	189 (19.8)
*Mean soda consumption*	*4.4 [5.9]*
**Psychosocial Community Characteristics**	
** *Neighborhood risks and resources* **	
Perceived neighborhood violence	
Low violence	364 (38.2)
Intermediate violence	283 (29.7)
High violence	307 (32.2)
Park access	
Has park access	774 (81.1)
Does not have park access	180 (18.9)
Mode of transportation to the nearest grocery store	
Car	775 (81.2)
Bus	38 (4.0)
Walking	123 (12.9)
Other	18 (1.9)
*Mean average number of miles traveled to the nearest grocery store*	*4.1 [6.1]*
*Mean community-level economic hardship*	*50.4 [17.5]*
** *Sense of community* **	
*Mean perceived community-level collective efficacy*	*33.8 [7.9]*
Neighborhood satisfaction	
Very satisfied/satisfied	852 (89.3)
Very dissatisfied/dissatisfied	102 (10.7)
**Sociodemographic Characteristics**	
Sex	
Female	472 (49.5)
Male	482 (50.5)
Age (years)	
18–30	401 (42.0)
31–40	245 (25.7)
41–50	127 (13.3)
Older than 50	181 (19.0)
Race/ethnicity	
Hispanic	468 (49.1)
Black	88 (9.2)
White	251 (26.3)
ANHOPI	147 (15.4)
Nativity Status	
Born in Los Angeles County	637 (66.8)
Native born but outside of Los Angeles County	203 (21.3)
Foreign born	114 (12.0)
Language spoken at home	
English	728 (76.3)
Not English	226 (23.7)
Education	
High school or less	184 (19.3)
Technical/vocational school or some college	343 (36.0)
College graduate/postgraduate	427 (44.8)
Employment Status	
Employed–full time	531 (55.7)
Employed–part time	114 (12.0)
Unemployed	104 (10.9)
Other employment status	205 (21.5)
Income	
Under $50,000	436 (45.7)
$50,000–$99,000	297 (31.1)
$100,000 or more	221 (23.2)
Marital Status	
Married	367 (38.5)
Single	511 (53.6)
Divorced/Separated/Widowed	76 (8.0)
*Mean number of children in the household*	0.8 [1.1]

^a^ Data were collected from a 2014 internet panel survey of Los Angeles County residents. This survey has been described elsewhere [38,39]. The number of cases and percentage may not add up to the total or 100%, respectively, due to rounding and missing values.

**Table 2 ijerph-18-09868-t002:** Associations between psychosocial community characteristics (PCCs) and fruit and vegetable and soda consumption, before and after controlling for sociodemographic characteristics: Results from an internet panel survey of Los Angeles County residents, 2014.

	Fruit and Vegetable Consumption		Soda Consumption	
	PCCs Only	Full Model ^a^	PCCs Only	Full Model^a^
	IRR (95% CI)	IRR (95% CI)	IRR (95% CI)	IRR (95% CI)
**Psychosocial Community Characteristics**				
Perceived neighborhood violence (ref: low violence)				
Intermediate violence	1.09 (0.98–1.23)	1.08 (0.97–1.20)	1.07 (0.87–1.33)	1.02 (0.82–1.25)
High violence	1.27 (1.13–1.43) ***	1.24 (1.11–1.40) ***	1.42 (1.14–1.77) **	1.24 (0.99–1.54)
Park access (ref: has park access)				
Does not have park access	0.95 (0.84–1.07)	0.94 (0.83–1.06)	0.89 (0.71–1.10)	0.88 (0.70–1.10)
Mode of transportation (ref: Car)				
Bus	1.18 (0.92–1.53)	1.25 (0.97–1.61)	1.47 (0.98–2.22)	1.53 (0.98–2.38)
Walking	1.03 (0.89–1.19)	1.08 (0.94–1.24)	0.99 (0.76–1.28)	0.89 (0.69–1.15)
Other	1.45 (1.02–2.06) *	1.38 (0.99–1.93)	1.24 (0.85–1.80)	1.20 (0.78–1.85)
Grocery store distance	1.01 (1.00–1.02) ^b^ **	1.01 (1.00–1.02) ^b^ *	1.02 (1.01–1.03) **	1.01 (1.00–1.02)
Community-level economic hardship	1.00 (1.00–1.00)	1.00 (1.00–1.00)	1.00 (1.00–1.01)	1.00 (0.99–1.00)
Perceived community-level collective efficacy	1.02 (1.01–1.02) ***	1.02 (1.01–1.02) ***	1.01 (1.00–1.02)	1.01 (0.99–1.02)
Neighborhood satisfaction (ref: very satisfied/satisfied)				
Very dissatisfied/dissatisfied	0.86 (0.74–1.01)	0.88 (0.76–1.03)	0.94 (0.69–1.29)	0.97 (0.71–1.33)

Note: IRR = Incidence rate ratio. ^a^ The full model adjusts for sociodemographic characteristics (data not shown). ^b^ The 95% confidence interval includes 1.00 with rounding. * *p* < 0.05, ** *p* < 0.01, *** *p* < 0.001.

## Data Availability

Available from the authors if data request is appropriate and feasible.

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
