# Peer review of "What Are the Relationships between Psychosocial Community Characteristics and Dietary Behaviors in a Racially/Ethnically Diverse Urban Population in Los Angeles County?"

_ijerph, 2021, doi:10.3390/ijerph18189868_

Round 1

Reviewer 1 Report

I enjoyed reading and learned new knowledge from your manuscript. Thank you.

As a reader and peer reviewer, I left a few questions for possible revision below. 

The first is the conceptual framework. The socio-ecological model describes an individual's habits/hobbies/behaviors from interpersonal, organizational, and communal relationships in society. Then how do the biopsychosocial model and environmental affordances model differ? Could you give us more explicit details? I still think that neighborhood risks and resources and sense of community in psychosocial community characteristics (PCCs) are not much different from the socio-ecological model. Does your model consider more community characteristics? Are they your unit of analysis? Would you please state that clearly?

Do you have research questions? Do you have a particular hypothesis you want to prove?

Seemingly preparatory work explored factors causing healthy fruit and vegetable (F+V) eating or unhealthy soda drinking. Let us consider a covariate and an independent variable. Of course, both covariates and independent variables in PCCs could affect eating and drinking behaviors, but which could affect more, either education or perceived neighborhood violence? How did PCCs play explaining dietary preferences independently?

In Table 1, you listed summary statistics. A few variables are highly skewed. For instance, many residents (80%) could access parks. Many more residents (81%) drove vehicles to the nearest grocery stores. About 89% were satisfied with the neighborhood's sense of satisfaction. Then my question is how they play a role of independent variables?    

I would not point out the limitation of negative binomial regressions using cross-sectional data instead of panel data or repeated cross-sectional data. (I appreciate that you wrote that in Section 4.3. Study Limitations. But I suggest more careful considerations to conclude your manuscript, considering your study merely captures a snapshot of community characteristics. I have other questions related to that. What statistical program/package did you use? Did you go over goodness-of-fit tests or other post-estimation tests?      

You listed many possibilities in 4.1. F+V Consumption and 4.2. Soda Consumption. First in 4.1., I intuitively agree on much that culture could influence F+V eating habits, likewise Soda Consumption. I also believe individuals in a community could have a particular eating/drinking habit. But I do not follow your linkage between perceived neighborhood violence and F+V consumption by race/ethnicity. According to Table 2, high violence increased the likelihood of not only F+V consumption but soda consumption. I am not entirely persuaded racial differences in high perceived neighborhood violence consumed more or less F+V. Instead, it is more plausible that respondents who live farther distances to the grocery store drink more soda because soda has a longer shelf life than F+V. Although you explained not many studies considered PCCs on F+V or soda consumption, please try to find as many as possible to make your findings robust.   

Now, what are your implications?  What do you suggest to policymakers and public health experts? Would you please elaborate on them for future studies and interventions?  

Author Response

Dear Reviewer 1,

We very much appreciate you taking the time to review our manuscript. The feedback that you provided helped us significantly improve it. We have included a response to each of your comments. Please see the attachment.

Some major changes that we made include redesigning the conceptual model to more adequately represent the pathways examine in our manuscript. We also included an analytic strategy section to better delineate the analyses carried out.

We hope that the changes made to the manuscript address your comments.

Reviewer 2 Report

Overall, this is a well-written paper that explains the relationship between environment and certain dietary behaviors. There were just a few items that the reviewer felt needed to be further expanded on. Please note the English/grammar were fine, but the only option is 'minor spell check required':

Title: Suggest including the region that this study was conducted in and that this was conducted in adults

Abstracts: Suggest including the p-values for those values that were statistically significant; specify the number of adults in the original survey analysis. Recommend that the conclusive statement also reflect implications for health professionals, researchers, etc.

Introduction: Recommend there is a bit more information about diversity and health disparity and how the environment impacts dietary intake. Explain why the focus is on F/V and soda consumption as opposed to salt, fat and other dietary behaviors that, at least in the literature, is an issue among certain racial/ethnic groups and SES. In the introduction mentioned the SEM, but yet have different models to guide this study, suggest pointing out issues with the SEM and why other models may be better to assess relationships between environment and dietary behaviors.

Methods: Explain if this was a one-time only survey or if this was to be conducted every 10 years, etc. As this survey was provided to a racially diverse population, no other languages was available? Clarify. Mentioned target quotas, expand on what was the power/sample size estimates.

Author Response

Dear Reviewer 2,
We thank you for your insightful feedback, which we believe helped to strengthen our manuscript. Please see the attachment, as it provides a response to each of your comments. 

Per your and other reviewer feedback, major changes that we made include redesigning the conceptual model to more adequately represent the pathways examine in our manuscript.  An analytic strategy section was also included to better delineate the analyses carried out.

Reviewer 3 Report

The article “Psychosocial community characteristics and dietary decisions: Implications for fruit, vegetable, and soda consumption in a racially/ethnically diverse urban population” by Robles, Kuo, and Tobin examined the associations between several measures of psychosocial community characteristics (PCCs) and diet behavior in adults who participated in the Los Angeles County (LAC) internet panel survey. The manuscript utilizes unique measures to further knowledge in the field of built environment research. The results would be of interest to other researchers in this area. However, several revisions should be addressed by the authors before publication.

Major comments

Introduction

  1. The authors mentioned on lines 51-52 that, other factors such as “emotions/psychological distress, neighborhood safety, and other aspects of community contexts” can motivate dietary decisions. There needs to be references for these statements. Additionally, it would be beneficial to provide examples of what “community contexts” mean.

  1. While the moderation analysis by race/ethnic groups was interesting, the reasoning for this analysis needs to be clarified. The authors mentioned that race and racism have resulted in differences in access to community-level resources and PCCs (lines 66-67). However, it is unclear how race may play a part in influencing how PCCs relate to dietary behaviors. One or two examples on this point would greatly strengthen this component of the manuscript.

  1. It is unclear how the authors ended up with the list of variables to be studied. For example, “community-level economic hardship”, “mode of transportation”, “neighborhood satisfaction” was not mentioned in the introduction at all. How may these factors relate to dietary behaviors? Relatedly, it would be easier for readers if the authors introduce the two domains of PCCs early on and structure the introduction section according to the domains.

  1. Several improvements can be made to Figure 1. (line 91-93).
    1. Clarify what the arrows mean (are they causal arrows, are they association arrows, is there meaning in the direction?).
    2. Currently, the figure suggests that covariates determine PCCs which influences dietary behaviors. Covariates also influence dietary behaviors through other means not related to PCCs. However, the author hypothesized that race/ethnicity may influence the relationship between PCCs and diet. Thus, there should be another arrow indicating this (e.g., from covariates box to the arrow between PCCs and diet).

Methods and Materials

  1. Starting in line 134, F+V consumption was described as a categorical variable. There needs to be clarification on how F+V consumption was used for the negative binomial regression.

  1. Same comment as above for the soda consumption.

  1. In general, there should be a separate statistical analysis subsection in the Methods and Materials section to talk about the negative binomial regression and moderation analysis. These terms were first seen in the abstract and second in the results section. They should be explained in the Methods and Materials section. In this subsection, the authors can describe how the binomial regression analysis was run, what is meant by the moderation analysis, and what software was used to analyze data.

  1. Lines 207-209, the authors can expand a little bit more why a factor analysis was conducted to create the composite perceived collective efficacy. Also, provide some definitions or examples of what is meant by “overlapping constructs”.

  1. Some of the covariates are coded out of order e.g., education was categorized as 0=college graduate/postgraduate; 1=high school education or less; and 2=technical/vocational school or some college. Category 2 and 1 should be switched if they are meant be ordered from highest to lowest. Income is also coded as 0=over $100,000; 1= <$50,000, and 2=$50,000-99,000. Category 2 and 1 should be switched. Please check that this did not influence the results, both the regression analyses and the descriptive tables.

Results

  1. The moderation analysis subsection can be improved significantly:
    1. As a general comment, typically, in moderation analysis, one would observe whether trends of the outcome of interest across levels of independent variables change depending on the moderator variable (e.g., is there a difference in how F+V consumption changes as neighborhood violence increases across different race/ethnic groups). In this manuscript, the authors only examined differences of the outcomes of interest at different levels of the independent variable between each category of the moderator variable (i.e., is there a difference in F+V consumption when individuals of different race/ethnic groups are exposed to high levels of neighborhood violence). This latter case is often referred to as stratified analyses instead. To prevent confusion, the authors should clarify what they mean by ‘moderation analysis’ in the Methods and Materials section.
    2. Provide units on any statistics present e.g., line 308 “… (White: 0.71, CI=0.55-0.92; ...) are these number of servings? Or are they probability?
    3. Lines 321-323, is this result accurate? From the figure, it seems that consumption of soda is lower in White participants compared to Hispanic counterparts when their primary mode of transportation is walking. Additionally, should there be a mention regarding white participants who travel by bus compared to Hispanic?
    4. Was there a reason why comparisons were only done against Hispanic participants vs. every other category?

Discussion

  1. This is not mentioned in the discussion, but it is curious that the mean F+V consumed is 5.6 servings per day. This is quite high, especially knowing that in general the US population is not meeting the recommended amount of F+V consumption. How does this amount (and amount of soda consumed) compare to other studies with adult population? Is it possible that the survey participants have better dietary intake than the general population? How may this affect interpretation of the results?

  1. Lines 386-390, is this only true for Black and Hispanic individuals? Are there studies showing that something similar is not observed for White or Asian and Native Hawaiian/Pacific Islander individuals?

  1. Lines 415; same comment as 10.c is this result accurate?

  1. F+V consumption was self-reported and this should be added as a limitation. Even if participants were given examples of serving size, over reporting of healthy foods and under reporting of unhealthy foods (social desirability bias) is common. This influences the accuracy of this measure and should be noted as a limitation.

  1. What are some strengths of the study?

Minor comments

  1. There is an issue with formatting, page numbers should be on the right side rather than the left according to the journal’s template. The line numbers on the first page (27-44) overlaps with the publisher’s box.

  1. On line 138, there’s a full stop that’s in red font rather than black.

  1. On line 149, worse consumption is defined as both 7 or more sodas per week and 1 or more sodas per week. Please clarify.

  1. Typo on line 229. Income category 0 should read $100,000 rather than $100,0000.

  1. Lines 233-247, can the authors comment on whether the demographic characteristics of the survey participants are reflective of LAC population in general?

  1. Line 257, results should read 24% rather than 25%.

  1. Table 2 footnotes, the authors should remove information about significant sociodemographic characteristics, since these are not presented on the table and already described elsewhere.

  1. On line 310, should it read “These relationships are depicted in Figure 2.”? Not figure 1.

  1. The contents in the leftmost column of tables need to have their alignment fixed. Currently the text starts and ends in different places and it’s hard to see subcategories vs. main categories. For example, “Perceived neighborhood violence” should be aligned with “Park access”. Another example, “Low violence”, “Intermediate violence”, and “High violence” should be aligned the same way and indented appropriately. It might be easier to just left-align the leftmost column.

Author Response

Dear Reviewer 3,

You provided excellent feedback, which we much appreciate. It has helped to significantly improve our manuscript. Please see the attachment for a response to each of your comments

We made major changes to the text and figure based on your and other reviewer comments. Some major changes include redesigning the conceptual model to more adequately represent the pathways examine in our manuscript, and including an analytic strategy section to better delineate the analyses carried out.

Round 2

Reviewer 3 Report

Thank you for making extensive revisions based on the comments. I believe that all comments have been addressed sufficiently.